# Co-Expression of Immunohistochemical Markers MRP2, CXCR4, and PD-L1 in Gallbladder Tumors Is Associated with Prolonged Patient Survival

**DOI:** 10.3390/cancers15133440

**Published:** 2023-06-30

**Authors:** Andrés Tittarelli, Omar Barría, Evy Sanders, Anna Bergqvist, Daniel Uribe Brange, Mabel Vidal, María Alejandra Gleisner, Jorge Ramón Vergara, Ignacio Niechi, Iván Flores, Cristián Pereda, Cristian Carrasco, Claudia Quezada-Monrás, Flavio Salazar-Onfray

**Affiliations:** 1Programa Institucional de Fomento a la Investigación, Desarrollo e Innovación, Universidad Tecnológica Metropolitana, Santiago 8940577, Chile; atittarelli@utem.cl; 2Millennium Institute on Immunology and Immunotherapy, Santiago 8380453, Chile; omar.barria@ug.uchile.cl (O.B.); e.sanders@lumicks.com (E.S.); nat13ab1@student.lu.se (A.B.); maria.gleisner@uchile.cl (M.A.G.); iflores@docente.uss.cl (I.F.); cristian.pereda@uchile.cl (C.P.); 3Disciplinary Program of Immunology, Institute of Biomedical Sciences, Faculty of Medicine, Universidad de Chile, Santiago 8380453, Chile; 4Laboratorio de Biología Tumoral, Instituto de Bioquímica y Microbiología, Universidad Austral de Chile, Valdivia 5090000, Chile; duribe@idibell.cat (D.U.B.); ignacio.niechi@uach.cl (I.N.); 5Molecular and Traslational Immunology Laboratory, Department of Clinical Biochemistry and Immunology, Pharmacy Faculty, Universidad de Concepción, Concepción 4070386, Chile; mabel.vidal@gmail.com; 6Computer Science Department, Universidad de Concepción, Concepción 4070386, Chile; 7Departamento de Informática y Computación, Universidad Tecnológica Metropolitana, Santiago 7800002, Chile; jorgever@utem.cl; 8Subdepartamento de Anatomía Patológica, Hospital Base de Valdivia, Valdivia 5090000, Chile; cristian.carrascohv@redsalud.gov.cl; 9Department of Medicine Solna, Karolinska Institute, 17176 Stockholm, Sweden

**Keywords:** gallbladder cancer, tumor markers, immunohistochemistry, tissue microarray, immune tumor microenvironment, multidrug resistance

## Abstract

**Simple Summary:**

Gallbladder cancer (GBC) is a significant health problem in Asia and Latin America, particularly in Chile. The limited therapeutic options for GBC demand the identification of targetable proteins with predictive value. We evaluated the prognosis impact of 18 different protein markers in primary lesions from 241 GBC patients using automated immunohistochemistry, a semi-automatic method for image analysis, univariate and multivariate statistical analyses, and machine learning algorithms. Our results show a significant association between the expression of the MRP2/CXCR4/PD-L1 cluster of markers and prolonged patient survival. Hence, our results suggest that this cluster could be a valuable prognostic tool for GBC.

**Abstract:**

Gallbladder cancer (GBC) is a rare pathology in Western countries. However, it constitutes a relevant health problem in Asia and Latin America, with a high mortality in middle-aged Chilean women. The limited therapeutic options for GBC require the identification of targetable proteins with prognostic value for improving clinical management support. We evaluated the expression of targetable proteins, including three epithelial tumor markers, four proteins associated with multidrug and apoptosis resistance, and eleven immunological markers in 241 primary gallbladder adenocarcinomas. We investigated correlations between tumor marker expression, the primary tumor staging, and GBC patients’ survival using automated immunohistochemistry, a semi-automatic method for image analysis, univariate and multivariate statistical analyses, and machine learning algorithms. Our data show a significant association between the expression of MRP2 (*p* = 0.0028), CXCR4 (*p* = 0.0423), and PD-L1 (*p* = 0.0264), and a better prognosis for patients with late-stage primary tumors. The expression of the MRP2/CXCR4/PD-L1 cluster of markers discriminates among short-, medium-, and long-term patient survival, with an ROC of significant prognostic value (AUC = 0.85, *p* = 0.0012). Moreover, a high MRP2/CXCR4/PD-L1 co-expression is associated with increased survival time (30 vs. 6 months, *p* = 0.0025) in GBC patients, regardless of tumor stage. Hence, our results suggest that the MRP2/CXCR4/PD-L1 cluster could potentially be a prognostic marker for GBC.

## 1. Introduction

Gallbladder cancer (GBC) is a highly aggressive and fatal biliary tract carcinoma (BTC). Unfortunately, it is often diagnosed in advanced stages due to nonspecific clinical symptoms, resulting in less than 10% of patients being candidates for surgical resection. The 5-year survival rate for GBC ranges from 4% to 60%, depending mainly on the disease stage at detection [1]. GBC incidence varies widely, with the highest cases reported yearly in some Eastern European regions, Asia, and Latin America. Chile has the world’s highest incidence and mortality rates [1,2,3].

Patients with advanced GBC have an average overall survival (OS) time of 4–14 months. Even with surgical resection, most progress to a metastatic stage highly resistant to conventional chemotherapy [4,5]. Although palliative systemic therapy and surgery combined with adjuvant chemotherapy are currently the best options for improving clinical outcomes for GBC patients, they have not been sufficient to increase patient survival rates significantly [6]. In this context, the overexpression of multidrug resistance-associated proteins (MRPs) in cancer cells is one of the leading causes of the intrinsic phenotype of chemotherapy resistance, including in GBC [7]. 

Molecularly targeted therapies and immunotherapies are among the most promising strategies for treating GBC [8]. Molecularly targeted therapies, commonly designed based on data from next-generation genome sequencing, are essential for treating many tumors. To date, specific biomarkers for GBC need to be better defined. Some studies have recognized different molecular characteristics of GBC compared to other forms of BTC. The top four genomic alterations in GBC are tumor protein (TP) 53 (59%), cyclin-dependent kinase inhibitor (CDKN) 2A/B (19%), AT-rich interaction domain (ARID) 1A (13%), and Erb-B2 receptor tyrosine kinase (ERBB) 2 (16%). GBC is also characterized by a high prevalence of amplification of the epidermal growth factor receptor (EGFR) 2 and by activator mutations in the Kirsten rat sarcoma viral (KRAS) gene [9,10,11]. 

Recently, immunotherapies based on immune checkpoint inhibitors (ICIs) targeting inhibitory pathways, such as cytotoxic T lymphocyte-associated protein 4 (CTLA-4) and programmed cell death protein-1 (PD-1) and its ligand (PD-L1), have shown promising results in BTC [12,13,14,15]. Moreover, combinations of ICIs with other molecular targets are being intensively tested in BTC [8]. Dendritic cell-based cancer vaccines using peptides or tumor cell lysates as a source of antigens [16,17,18], used alone or combined with other therapies, constitute a feasible and promising strategy. 

GBC has a multifactorial etiology, with various risk factors contributing to its development and aggressiveness, including gender, genetic-related factors, parasitic infections, smoking, alcohol consumption, chronic inflammation (cholecystitis), and gallstones (cholelithiasis), among others [19,20]. From a biomolecular point of view, multiple tumor, genetic, inflammatory, and drug-resistance markers associated with BTC could impact the course of the disease [21]. 

In this respect, the wide range of risk factors and the failure of current therapies have highlighted the need to identify new biomarkers for the early detection, prognostic stratification, or therapeutic management of GBC patients. In this vein, proteomic and genomic studies of tumor samples and cell lines have been used to elucidate the role of potential prognostic markers [22,23]. Although the immunohistochemical (IHC) marker pattern of GBC is not entirely specific, it shows a similarity to that observed in bile duct and pancreatic carcinomas, and has been helpful for the prognostic stratification of GBC patients [24].

Here, we evaluate the expression of targetable proteins, including three epithelial tumor markers, four proteins associated with multidrug and apoptosis resistance, and eleven immunological markers in 241 primary gallbladder adenocarcinomas. We evaluate the correlations between tumor marker expression, primary tumor staging, and GBC patients’ survival using automated IHC in tissue microarrays (TMA), a semi-automatic method for image analysis, univariate and multivariate statistical analyses, and machine learning algorithms. Our results identify different molecular markers obtained using univariate and multivariate analysis, which may impact disease prognosis and management. 

## 2. Materials and Methods

### 2.1. Patient Biopsies

Cholecystectomy specimens from 2000 to 2019 were obtained from the pathological anatomy subdepartment of Hospital Base Valdivia (Valdivia, Chile). The tumor samples were categorized using code C.23, according to the International Classification of Diseases for Oncology. The selection criteria for the biopsies were based on sample etiology; thus, only invasive primary gallbladder adenocarcinomas were selected. The pathologist excluded squamous carcinomas, neuroendocrine carcinomas, tumor-infiltrated lymph nodes, and metastases. According to these criteria, we collected 241 biopsies distributed in 9 arrays per marker, with around 29 patients each. Primary tumor staging was established based on the reported imaging findings by a pathologist who evaluated the resected specimens. The GBC samples were classified into two categories: primary tumors in an early stage, defined by adenocarcinoma confined to the tunica muscularis (tumor in situ (TIS), T1a, or T1b), or late-stage tumors encompassing T2, T3, and T4, indicating a more advanced tumor progression, according to the eighth edition of the American Joint Committee on Cancer (AJCC) Staging Manual and the AJCC/UICC TNM Classification 2010 [25]. This retrospective study was performed in agreement with the Code of Ethics of the World Medical Association (Declaration of Helsinki), printed in the British Medical Journal (18 July 1964), and approved by the Bioethical Committee for Human Research of Valdivia Regional Hospital (protocol code 403, date of approval December 2015) and the Universidad de Chile Ethics Committee for studies involving humans (code 086-2017, 27 June 2017). The Bioethical Committee for Human Research of Valdivia Regional Hospital (code 082-2020, 23 March 2020) authorized the inclusion of samples until 2019. All patients signed a letter of informed consent at the time of surgery.

### 2.2. Tissue Microarrays

The Quick-Ray UT06 Manual Tissue Microarrayer (Unitma Co., Ltd., Seoul, Republic of Korea) was used following the manufacturer’s instructions. A 2 mm puncher tip, and its corresponding puncher, and receptor blocks with 2 mm × 60 mm perforations were utilized. This procedure allowed for the precise localization of the chosen site when the tissue slices were positioned against their respective donor block (the tissue was studied in a paraffin block). Before tissue extraction, the donor blocks were incubated at 37–40 °C for 15–20 min. Subsequently, the samples were inserted into the puncher tip to a depth of 5 mm (per needle measurement), resulting in a tissue cylinder deposited onto a receptor block.

The block was positioned on a horizontal and level surface, and the needle of the puncher tip was aligned perpendicularly with the holes. Tissue was slowly injected into the corresponding hole to form a tissue cylinder. Subsequently, the cylinders were gently pressed to create an even surface on the top of the receptor block. In parallel, a grid sheet was created to record the position of each tissue piece and its corresponding case number. The receptor blocks used for the microarrays had 60 cylinder perforations, but only 59 were utilized each time, leaving one strategically empty for individualization and orientation to facilitate further identification.

The receptor block was then placed face down on a base mold (from the kit) and incubated at 70 °C for 30–60 min until the blocks became utterly transparent. Once transparent, they were embedded in an inclusion cassette and allowed to solidify on a cold plaque. Finally, a rotary microtome was employed to produce slices with a width of 3 microns. They were then mounted on slides measuring 75 mm × 25 mm, along with their respective positive tissue controls. The positive control tissues were: gastric cancer for MUC1; colon cancer for CEA, CA19-9, and survivin; fetal liver for MRP2; adult liver for MRP3 and Pg-P; and tonsil for the rest of the markers analyzed. 

### 2.3. Immunohistochemistry

IHC was performed as previously described [24]. Briefly, the samples were processed using two different pieces of equipment. The Tissue-Tek Prisma Plus Automated Slide Stainer machine (Sakura Finetek USA, Torrance, CA, USA) assessed the tissue morphology with hematoxylin and eosin staining. For staining with chromogen 3,3-diaminobenzidine (DAB), which generates the color for measuring the marker expression, Benchmark GX equipment was used, which enabled the processing of 25 slides simultaneously. Antigenic retrieval was conducted for 30 min at 95 °C, followed by incubation with the primary antibody for 1 h. Subsequently, the secondary antibody and ultraviolet Universal DAB Detection Kit (Roche, Basel, Switzerland) were added, and all the kit reagents were incubated for 1 h to ensure proper staining. 

Sample dehydration was meticulously performed manually, involving immersion in plastic wells containing ascending gradients of 70%, 95%, and 99% ethanol for 5 min each. Subsequently, the samples were washed in xylene for 10 min to complete the dehydration process. Finally, the slides were carefully mounted by sealing them with a coverslip, utilizing the Entellan reagent for optimal storage and subsequent analysis. This meticulous approach aimed to ensure the proper preservation and protection of the samples for accurate and reliable results during further analysis.

### 2.4. Antibodies

The mouse monoclonal antibodies used in this study were: anti-MUC1 (clone BSB-44, Bio SB, dilution 1:50), anti-CEA (clone BSB-31, Bio SB, prediluted), anti-CA19-9 (clone 121SLE, dilution 1:75), anti-MRP2 (clone M2III-6, dilution 1:100), anti-MRP3 (clone M3II-9, dilution 1:5), anti-Pg-P (clone JSB-1, dilution 1:5), anti-TIM3 (clone TIM3/3113, dilution 1:50), anti-PD-L1 (clone ABM4E54, dilution 1:500), anti-HLA-ABC (clone EMR8-5, dilution 1:50), anti-IL-8 (clone 807, dilution 1:50) (Abcam, Cambridge, UK), anti-survivin (clone 8E2, dilution 1:30), anti-TNF-a (clone F6C5, dilution 1:50), and anti-TGF-b (clone TB21, dilution 1:50) (ThermoFisher Scientific, Waltham, MA, USA). The rabbit polyclonal antibodies used were: anti-A2aR (PA-33323, ThermoFisher Scientific, dilution 1:200), anti-IFN-g (ab25101, dilution 1:500), anti-CXCR4 (p-S339) (ab74012, dilution 1:75), anti-CCR7 (ab140758, dilution 14 mg/mL), and anti-IL-6 (ab6672, dilution 1:600) (from Abcam).

### 2.5. Determination of Immunoreactive Scores 

The NanoZoomer 2.0-HT machine (Hamamatsu Photonics, Shizuoka, Japan) was used for digitization and the NDP.view 2.5 software (Hamamatsu Photonics, Shizuoka, Japan) for imaging visualization. Staining mark detection was analyzed using Qupath [26], a pathologists-validated software based on a Java interface that is free to use.

The expression levels for each marker were defined by the immunoreactive score (IRS), which ranges from 0 to 12 (or from 0 to 9), and is calculated by multiplying the percentage score of positive cells (0–4) with the intensity of staining (0–3). The intensity score was determined by calculating the optical density ratio to pixels in the sample, resulting in a constant value that was then transformed into one of four categorical scores (0, 1, 2, or 3). For the immune-associated markers, the percentage of positive cells in a tumor area was transformed into one of four (0–3) or five categorical scores (0–4), depending on the type of marker. For TIM3 and PD-L1, the percentages of positive cells were divided into four categorical scores, as previously described by Peng et al. (2017) [27] and Neyaz et al. (2018) [28], respectively. For A2aR, the percentages of positive cells were divided into five categorical scores, as described by Wu et al. (2019) [29]. For the rest of the immunological markers, the percentages of positive cells were divided evenly into five categorical scores (0–19%, 20–39%, 40–59%, 60–79%, and 80–100%). The non-immune-related markers analyzed (MRP2, MRP3, Pg-P, CEA, CA19-9, survivin, and MUC1) showed a homogeneous distribution of expression in the tumor tissue; therefore, their IRS only considered the intensity scores.

### 2.6. Statistical Analysis

Statistical analyses were performed using GraphPad Prism 9 (GraphPad Software Inc., San Diego, CA, USA). A chi-squared with Yates correction analysis was used to analyze the frequency of each group concerning the studied parameters. Kaplan–Meier and log-rank (Mantel–Cox) tests were used to construct and evaluate the OS data. Univariate and multivariate analyses were performed using the Cox proportional hazard regression model (Breslow method) to study the effects of different variables on OS. All the variables included in the univariate regression analysis were also considered in the multiple regression analysis except survivin, which exhibited homogeneous staining among the tumor samples. Receiver operating characteristic (ROC) curves were drawn to evaluate each marker’s predictive and cut-off values. Differences were considered statistically significant at *p* < 0.05.

For the global feature contribution, all the calculations were performed in Python 3.9. We used the SMOTE algorithm [30] from the imbalanced-learn library as an over-sampling method to reduce the imbalance class in the datasets. This algorithm increases the sensitivity of a classifier to the minority class. We applied machine learning to classify the patients into different groups to study. We used Scikit-learn [31] and the Random Forest Classifier algorithm. This strategy is an ensemble method, which is able to combine predictions through estimators. The hyperparameter search was performed using GridsearchCV. The data were split into training data (70%) and test data (30%). The feature importance analysis per group was examined using the Shapley Additive explanation algorithm [32], where the variables were ranked in descending order. 

A dimensionality reduction analysis was performed using uniform various approximation and projection (UMAP), a nonlinear method known for its fast execution time, consistency, significant cluster organization, and preservation of continua compared to other existing methods. A hyperparameter adjustment followed the approach described by McInnes et al. [33] as a reference. A grid search was conducted for the parameters of n_neighbors and min_dist, and the best combination was n_neighbors = 20 and min_dist = 1.

## 3. Results

### 3.1. Clinicopathological Characteristics of the Gallbladder Cancer Patient Cohort

Most of the GBC patients included in this study were women (191 vs. 50) aged 37–87 years (mean 65.1 ± 11.2). Using T stages that consider the size and spread of the tumor according to the AJCC/UICC TNM Classification 2010, we categorized the samples into two groups: patients with early stage tumors (TIS + T1) and patients with late-stage tumors (T2, T3, and T4). Among the included patients, we observed 22 (9%) with tumors in situ (TIS), 24 (10%) with T1, 52 (22%) with T2, 137 (57%) with T3, and 6 (2%) with T4. The mean age of the patients with early stage tumors was slightly but significantly lower than those with late-stage tumors (61.6 ± 11.4 vs. 65.9 ± 11, *p* = 0.0181) (Table 1). A timely diagnosis is the primary challenge for the management of this cancer. The median follow-up period was 145 months for the complete cohort, which was not statistically different between the patients with early stage (99.2 months) and late-stage tumors (146 months) (*p* = 0.47). In this study’s cohort, 90% of the male and 78.5% of the female patients were diagnosed in the late stages of the disease, although the difference between the two groups was not statistically significant. In line with the accumulated evidence, our study’s median OS time and five-year survival rate were significantly lower for the patients with late-stage tumors (10 months, 11.8%) than for the patients with early stage ones (220 months, 52.2%) (*p* < 0.0001). Accordingly, tumor differentiation status was inversely associated with patient OS time (6 vs. 178.5 months) and five-year survival (3.1% vs. 46.1, hazard ratio (HR) = 4.22) (*p* < 0.0001), when compared to poorly and well-differentiated tumors, respectively (Table 1). 

### 3.2. Immunohistochemical Evaluation and Immunoreactive Score Analysis of Multiple Markers in Gallbladder Cancer Tissues

We determined the IRS using an automated IHC for eighteen (18) markers in fixed tumor samples from GBC patients using open-source tools for digital image analysis (Figure 1). The analyzed proteins included: three epithelial tumor markers, mucin 1 (MUC1), carcinoembryonic antigen (CEA), and carbohydrate antigen 19-9 (CA19-9 or sialyl-Lewis); four multidrug or apoptosis resistance-associated markers, MRP2, MRP3, P-glycoprotein (Pg-P, also known as ABCB1), and survivin; and eleven immunological markers, T-cell immunoglobulin and mucin-domain containing-3 (TIM3), adenosine A2a receptor (A2aR), PD-L1, major histocompatibility class I (human leukocyte antigen (HLA)-ABC), interferon (IFN)-g, interleukin (IL)-8, C-X-C chemokine receptor type 4 (CXCR4), tumor necrosis factor (TNF)-a, CC-chemokine receptor 7 (CCR7), transforming growth factor (TGF)-b, and IL-6 (Figure 2 and Appendix A).

The distribution of the IHC tumor marker’s IRS according to the primary tumor staging is shown in Table 2. The IRS scale for the positive-stained tissue samples for MUC1, CEA, and CA19-9 was 1–3, considering the staining intensity scores, and we categorized IRS = 1 and 2 as low and IRS = 3 as high expression (Figure 2 shows representative CEA and MUC1 photographs). The IRS scale for the positive-stained tissue samples for MRP2, MRP3, and Pg-P was 1–4, considering the staining intensity scores, and we categorized IRS < 4 as low and IRS = 4 as high expression (Figure 2 shows representative MRP2 and MRP3 photographs). The IRS scale was binary for survivin, meaning negative or positive staining. For the immunological markers, the IRS scale for the positive-stained samples was 1–4, 6, and 9 for TIM3 and PD-L1, and 1–4, 6, 8, 9, and 12 for the rest, considering the staining intensity scores x positive cells proportion scores. For TIM3 and PD-L1, IRS < 6 was considered as low, while IRS = 6 and 9 was considered as high expression (Figure 2 shows representative PD-L1 images). For the rest of the immunological markers, we considered IRS < 9 as low and IRS = 9 and 12 as high expression (representative CXCR4 images are shown in Figure 2). Representative photographs for the rest of the tumor markers are shown in Appendix A.

Most of the analyzed markers showed a high positivity rate (>80% of the tissue samples were positive); only TIM3, TNF-α, and CCR7 showed a lower positivity range (65–80%) (Table 2). Although the percentage of positivity for CEA expression was similar between early stage and late-stage tumors (83.9 vs. 85.3%), the proportion of tumors with a high CEA expression was significantly higher for late-stage tumors compared to early stage ones (*p* = 0.0212) (Table 2, Figure 3A). For MRP3, TIM3, PD-L1, TNF-a, and IL-6, the proportion of tumors with a low IRS was significantly increased for late-stage than for early stage tumors (*p* = 0.0136, *p* = 0.003, *p* = 0.0205, *p* = 0.0077, and *p* = 0.0004, respectively) (Table 2, Figure 3A).

### 3.3. Association of Tumor Markers with Gallbladder Cancer Patient Prognosis 

We analyzed the association of tumor marker IRS levels with the survival of GBC patients. There was no statistically significant association between the expression levels of any of the 18 markers and the OS time of patients with early stage primary tumors (Appendix A). Therefore, unless stated, the results described below were obtained for patients with late-stage primary tumors. Patients with a high tumor expression of CEA have a worse prognosis than those with tumors with a negative/low expression of this marker (median OS 7.2 vs. 10.6 months, *p* = 0.0157; HR = 1.38, 95% confidence interval (CI) = 1.078–1.786, *p* = 0.0123) (Figure 3B,C). On the other hand, when compared to a negative/low tumor expression of MRP2, high levels of MRP2 correlated with a better prognosis (median OS 8 vs. 16.5 months, *p* = 0.0073; HR = 0.6745, 95% CI = 0.4756–0.9491, *p* = 0.0255) (Figure 3B,C). Similarly, a high tumor expression of TNF-α (median OS 7.4 vs. 16.2 months, *p* = 0.0033; HR = 0.6706, 95% CI = 0.5158–0.871, *p* = 0.0028) or CXCR4 (median OS 6.3 vs. 12 months, *p* = 0.0012; HR = 0.7221, 95% CI = 0.5855–0.8959, *p* = 0.0027) correlated with a better prognosis than negative/low tumor expressions (Figure 3B,C).

Despite the statistical significance observed for an association between tumor expression levels of CEA, MRP2, TNF-a, and CXCR4 with patients’ OS time (Kaplan–Meier and univariate Cox regressions), an ROC curve analysis indicated a poor prognostic value for these markers when independently analyzed, given that the areas under the curves (AUC) were <0.75 (Figure 3D), a cut-off value for the clinically discriminatory power of the test [34]. 

Therefore, we used multivariate statistical methods to evaluate the prognosis value of combined tumor markers. First, we used a classification machine learning model using a random forest classifier to evaluate the contribution of the global feature (tumor markers expression). In this model, patients were classified according to tumor staging (early and late-stage tumors) with a global accuracy of 90%. To understand the importance of the features, we applied the Tree Shapley additive explanation (SHAP) algorithm [35] to measure the contribution of each marker on the model output (Tree SHAP values) for individuals (patients) in the training dataset. The feature importance plot in Figure 4A (left) clarifies the differential importance of the variables (markers) we used to classify the patients according to tumor staging. The features were ordered by the absolute sum value of their effect magnitudes on the model, with MRP2 and CXCR4 being the most critical features (Figure 4A, left). In addition, as the Tree SHAP values are derived from an individualized model interpretation approach, we performed an individualized interpretation for each sample. The SHAP summary plots in Figure 4A (right) show how the contribution of an individual feature on the model output is affected by its value. The position of each dot is the impact of the feature, and the color of the dot represents the value of that feature.

Additionally, a multivariate Cox regression analysis indicates a significant association between the expression levels of MRP2 (HR = 0.4162, 95% CI = 0.2307–0.7306, *p* = 0.0028), CXCR4 (HR = 0.6261, 95% CI = 0.3952–0.9784, *p* = 0.0423), and PD-L1 (HR = 0.5849, 95% CI = 0.3623–0.9347, *p* = 0.0264), with a better prognosis for patients with late-stage tumors. Similarly, this analysis shows a small but significant association between MUC1 and IL-6 expression with a worse patient prognosis (HR = 1.783, 95% CI = 1.094–2.959, *p* = 0.0223; and HR = 1.847, 95% CI = 1.012–3.368, *p* = 0.0447, respectively) (Figure 4B, left). An analysis of the complete cohort shows that MRP2 (HR = 0.39, 95% CI = 0.2172–0.6992, *p* = 0.0018) and PD-L1 (HR = 0.6, 95% CI = 0.3739–0.9352, *p* = 0.0267) are associated with a better prognosis, whereas MUC1 expression (HR = 1.66, 95% CI = 1.03–2.751, *p* = 0.0415) is associated with a poor prognosis for GBC patients, regardless of tumor staging (Figure 4B, right). A dimensionality reduction analysis using UMAP shows that patients cluster according to their OS times (short-term, medium-term, and long-term survival) only when the expression levels of the MUC1/IL-6/MRP2/CXCR4/PD-L1 group of markers are used as variables. Any other random combination of five tumor markers did not allow for the discrimination among GBC patients according to their OS times (Figure 4C). Then we performed a serial UMAP analysis to detect the minimal number of markers in the cluster MUC1/IL-6/MRP2/CXCR4/PD-L1, whose combination could discriminate among patients according to their OS times. We found that the cluster MRP2/CXCR4/PD-L1 fit this criterion, while any other combination of three markers among MUC1/IL-6/MRP2/CXCR4/PD-L1 did not (Figure 4C). Based on these results, we further performed an ROC curve analysis on patients with late-stage tumors, which showed a good prognostic value for the high co-expression of MRP2, CXCR4, and PD-L1 (*n* = 10 patients), as compared to the negative/low co-expression (*n* = 40 patients) of these markers (AUC = 0.85, *p* = 0.0012) (Figure 4D, left). Consistently, high co-expression levels of the three markers combined were strongly associated with increased OS times compared to their negative/low co-expression (median OS 6 vs. 19 months, *p* = 0.007) (Figure 4E, left).

Among the 46 patients with early stage primary tumors, two met the criteria of having a negative/low co-expression of the cluster MRP2/CXCR4/PD-L1, and five of these patients had the opposite profile, which made it impossible to statistically evaluate its predictive value for patients with early stage tumors. However, when these seven patients were included in the total group of patients analyzed (total *n* = 57 patients), we found that a high co-expression of MRP2/CXCR4/PD-L1 had an excellent predictive value (AUC = 0.84, *p* = 0.0001, Figure 4D, right) of a favorable OS (30 months vs. 6 months, HR = 0.35, CI = 0.19–0.63, *p* = 0.0005), regardless of the tumor stage (Figure 4E, right). There were no statistically significant differences in the mean age, gender distribution, disease stage (stages I, II, III, and IV), resection (R) status, or tumor differentiation among the patients with a high co-expression or a neg/low co-expression of MRP2/CXCR4/PD-L1 (Table 3). However, the group with high MRP2/CXCR4/PD-L1 co-expression exhibited a higher proportion of early stage tumors than the group with neg/low co-expression (*p* = 0.0038, Table 3). Therefore, it is necessary to consider tumor staging as a potential confounding variable that could influence the interpretation of the impact of MRP2/CXCR4/PD-L1 co-expression on the overall patient cohort. However, there were no significant differences in the baseline variables among patients with late-stage tumors between those with high co-expression and neg/low co-expression of MRP2/CXCR4/PD-L1 (Table 3).

## 4. Discussion

GBC is the most common BTC and is highly lethal. Often, it is diagnosed at a late disease stage as an incidental finding in tissue specimens after cholecystectomy for gallstones. Screening and early diagnosis remain arduous because patients rarely have specific symptoms. Several risk factors are involved in GBC, such as gallstones, age, sex, ethnicity, gallbladder polyps, obesity, diabetes, metabolic syndrome, cholangitis, family history of gallstones, *Salmonella enterica* serovar Typhi, and diet chemical exposure [21].

The molecular pathology of GBC is characterized by a high prevalence of inactivating somatic mutations in TP53, which occurs in 40% of primary GBC, according to the Catalog of Somatic Mutations in Cancer (COSMIC) database. Other recurrent mutations, which occur in <25% of primary tumors, are found in ERBB2, ERBB3, phosphatidylinositol-4,5-bisphosphate 3-kinase catalytic subunit alpha, CDKN2A/B, ARID1A, ARID2, mothers against decapentaplegic homolog 4, E74-like ETS transcription factor 3, and KRAS; most of these may be deleterious or potentially pathogenic [9,36,37]. 

Targeted therapies have significantly advanced the treatment of BTC, with the approval of fibroblast growth factor receptor inhibitors, such as Infigratinib and Pemigatinib, as well as the isocitrate dehydrogenase-1 inhibitor Ivosidenib. However, these molecular targets are relatively rare in GBC. Some clinical trials have suggested that targeting the human epidermal receptor 2/neu may be a promising option for advanced GBC patients, either as a single agent or combined with systemic chemotherapy [38]. Other molecular targets currently being investigated in GBC include EGFR, vascular endothelial growth factor receptor, mitogen-activated protein kinase, and rapidly accelerated fibrosarcoma kinase [21].

From an immunological perspective, GBC generally exhibits an immunosuppressed tumor microenvironment (TME). Less than 50% of tumors demonstrate an infiltration of CD3+ T cells [39]. However, CD3+CD8+ tumor-infiltrating T lymphocytes (TIL) correlate with better survival, whereas CD4+Foxp3+ TIL is associated with a poorer prognosis [39]. Moreover, GBC tumors often exhibit high levels of PD1 and PD-L1 positivity, with some TILs expressing PD-L1. Despite this, ICIs have shown only modest efficacy as single-agent treatments for BTC, including GBC. The balance between CD8+ and Foxp3+ cells in the TME has been proposed as a potential predictor of survival among GBC patients [39]. Therefore, boosting natural antitumor immune responses through cancer vaccines could be an adjuvant treatment option for GBC patients.

The systematic investigation of molecular biomarkers at the tumor site could significantly contribute to our understanding of tumor biology and have potential implications for risk prediction and effective biotherapies design. For instance, in a study analyzing 17 gastrointestinal IHC tumor-associated markers, only four markers (MUC6, CK17, CD10, and Villin) were significantly associated with OS in patients with GBC [24]. Additionally, in both China, a medium-risk area, and Chile, a high-risk area, GBC was found to be correlated with several circulating chemokines and markers associated with neutrophils [40,41]. Furthermore, increased levels of IL-6 have been observed in patients with cholelithiasis. They were even higher in those with early stage tumors, suggesting that IL-6 levels could reflect the transition from a healthy state to cholelithiasis, and eventually to GBC [42].

In this study, we assessed the expression of targetable proteins, including epithelial tumor markers, proteins associated with multidrug and apoptosis resistance, and immunological/inflammatory markers, using automated IHC in TMA. We employed different algorithms and statistical methods to analyze the data. Identifying the molecular markers that correlate with prognosis or would help guide treatment decisions has proven challenging in many cases, particularly when the presence or absence of a marker is intended to be associated with a potential effect. The presence of molecular markers in tumor tissue may not be a cause but a reflection or product of an undetected multifactorial situation. In such cases, multivariate statistical analyses, such as those performed in our study, would allow for exploring associations without necessarily establishing cause-and-effect relationships. 

Our results show that none of the 18 IHC markers were statistically associated with the prognosis of patients with early stage primary tumors (TIS + T1), which may have been influenced by the limited number of these types of patients in our study (*n* = 46). Although all the markers were highly expressed (>65%) in both early and late-stage tumors, we observed variations in the intensity of marker expression between early and late-stage tumors for six markers. The proportion of tumors exhibiting high CEA expression was notably higher for late-stage than early stage primary tumors (Figure 3A). This observation aligns with the poorer prognosis for late-stage tumors in patients exhibiting high CEA expression (Figure 3). CEA is a glycoprotein anchored to the cell surface, initially identified in human colon cancer tissue extracts [43]. The ASCO guidelines recommend determining serum CEA as the preferred tumor marker for metastatic colorectal cancer [44]. While the tumor expression levels of CEA in our late-stage cohort did not exhibit a statistically significant prognostic value (based on an ROC analysis), our findings are in line with existing evidence that suggests elevated CEA serum levels are associated with a poor response to chemotherapy and shorter survival times for patients with colorectal cancer [45].

Among the eleven immunological markers examined, three (PD-L1, TNF-α, and IL-6) demonstrated a decline in the proportion of tumors with a high marker expression in late-stage compared to early stage tumor samples (Figure 3A). Univariate analysis revealed a potential association between TNF-α expression and improved survival times for GBC patients with late-stage tumors (HR = 0.67, *p* = 0.0028). However, further examination, including ROC and multivariate analysis, indicated that the prognostic value of tumor TNF-α expression was limited. Although, to our knowledge, this is the first study to propose a potential prognostic value for intratumor TNF-α in BTC, it is noteworthy that a retrospective multivariate analysis of 102 patients with colorectal cancer demonstrated an increased TNF-α concentration to be an independent prognostic factor (HR = 0.29). These findings support the potential use of intratumor TNF-α as an indicator of TIL function, and as a prognostic parameter in colorectal cancer [46].

The results obtained in this study may seem controversial in comparison to previously published results elsewhere, in which MRP2, CXCR4, and PD-L1 were reported to be associated with the poor prognosis of patients with various cancers when expressed at high levels. Indeed, contrary to previous studies which associate the expression of drug transporters, such as MRP2, with more aggressive behavior of GBC cells and a worse clinical prognosis [47], our findings suggest that the high expression of MRP2 may be associated with a better prognosis for GBC patients. In the cohort study conducted by Kim et al. [47], MRP2 expression was positive in 53.1% (76/143) of the GBC samples. In contrast, our cohort showed a much higher positivity rate of 97.4% (188/193); directly comparing positive and negative markers was not feasible. Notably, both studies utilized the same antibody and dilution for MRP2 detection (clone M2 III-6, Abcam, 1:100), which suggests a potential biological difference in tumor MRP2 expression levels between the Korean and Chilean cohorts. A possible interpretation of this potentially controversial result is that in our cohort, the high positivity rate of MRP2 could indicate a physiological role of MRP2, which supports the terminal excretion and detoxification of endogenous organic anions and the uptake/excretion of bile salts in the absence of chemotherapy [48]. It is worth noting that the patients included in our study had not undergone any pre-operative chemotherapy at the surgical time point. In another study carried out in Brazil among BTC patients, MRP2 positivity was observed in 92.3% of hepatocellular carcinoma and 96.3% of cholangiocarcinoma tumors [49]. These findings are more consistent with the results observed in our study, suggesting some degree of similarity between the Chilean and Brazilian cohorts, possibly attributable to their geographic and genetic proximity.

The CXCL12-CXCR4 axis, which has been implicated in promoting metastasis and therapy resistance, is a classic marker for poor prognosis in various solid tumors, including GBC [50,51]. Although our findings contradict numerous reports reviewed in [50], they are consistent with a study demonstrating CXCR4 expression as a favorable prognostic indicator for survival in multiple myeloma patients [52]. These controversial results are congruent with the fact that the clinical inhibition of the CXCL12-CXCR4 axis using FDA-approved inhibitors, such as AMD3100 or Plerixafor, has yielded inconsistent outcomes, contrary to promising preclinical findings [50].

A high expression of CXCR4 in the cytoplasm or membrane of tumor cells is often a predictor of a poor prognosis. In contrast, a high expression in the nucleus is typically associated with a better prognosis. For example, in breast cancer, CXCR4 was found in both the cytoplasm and nuclei of tumor cells, but only the cytoplasmic expression was linked to lymph node metastasis [53]. In this context, Su et al. [54] and Wagner et al. [55] discovered that plasma membrane expression of CXCR4 in lung adenocarcinoma is an independent risk factor associated with poor disease-free survival. In contrast, nuclear expression confers a survival benefit. Therefore, CXCR4 may promote tumor cell proliferation and metastasis when present in the cytoplasm or cell membrane. Still, its effects are likely to be less pronounced when it is localized in the nucleus. Although not analyzed, these antecedents suggest that CXCR4 could be preferentially accumulated in the nucleus of tumor cells in our cohort. Further research and clinical trials are warranted to understand better the complex role of CXCR4 (particularly the expression of the ligand-activated p S339 form of CXCR4) in cancer progression and to explore potential strategies for the effective therapeutic targeting of this axis in different tumor types, including GBC.

While PD-L1 expression in tumor cells is associated with a favorable prognosis for various cancer types, such as lung cancer [56,57] and ovarian cancer [58], a study of 174 GBC patients showed that OS did not correlate with tumor PD-L1 expression [28]. Our univariate analysis of PD-L1 is consistent with these findings; however, we observed a potential association with prognosis when analyzed jointly with CXCR4 and MRP2 co-expression (Figure 4C–E). Of note, many reports have also shown an association between PD-L1 expression and poor prognosis for patients with different solid tumors, including hepatocellular carcinoma [59], breast cancer [60], and oral squamous cell carcinoma [61]. The differential immune landscape status of the different cohorts studied could explain this disagreement. As anticipated, our findings demonstrate that a multivariate analysis yields markers with higher clinical significance than a univariate analysis. UMAP analysis revealed that the co-expression levels of a cluster of three markers (MRP2/CXCR4/PD-L1) could effectively discriminate between short-term, medium-term, and long-term survival for GBC patients with late-stage tumors. We found that among our late-stage tumor cohort, 40 patients met the criteria of having negative/low tumor co-expression of MRP2/CXCR4/PD-L1 which, according to our results, translates into an inferior prognosis (HR = 2.62, median OS time of only six months).

Moreover, high MRP2/CXCR4/PD-L1 co-expression could have a prognostic value for GBC regardless of tumor staging (right panels of Figure 4D,E). Furthermore, ROC curve analysis revealed an excellent prognostic value for this marker cluster, with an AUC of 0.85 (*p* = 0.0012). Although further validation using an independent cohort of GBC patients is warranted, our results suggest that these markers have the potential to serve as targets for a prognostic tool.

## 5. Conclusions

In conclusion, our multivariate analysis revealed a significant association between MRP2, CXCR4, and PD-L1 co-expression and an improved prognosis for GBC patients, regardless of tumor staging. The combined expression of MRP2, CXCR4, and PD-L1 showed discriminatory power in distinguishing among short, medium, and long-term survival, with a significant prognostic value demonstrated by the ROC analysis (AUC = 0.85, *p* = 0.0012). The high expression of this cluster of markers was associated with increased overall survival times, suggesting that MRP2, CXCR4, and PD-L1 may serve as valuable prognostic markers for GBC. Further studies are warranted to validate these findings and explore their potential as clinical management support tools for GBC patients.

## Figures and Tables

**Figure 1 cancers-15-03440-f001:**
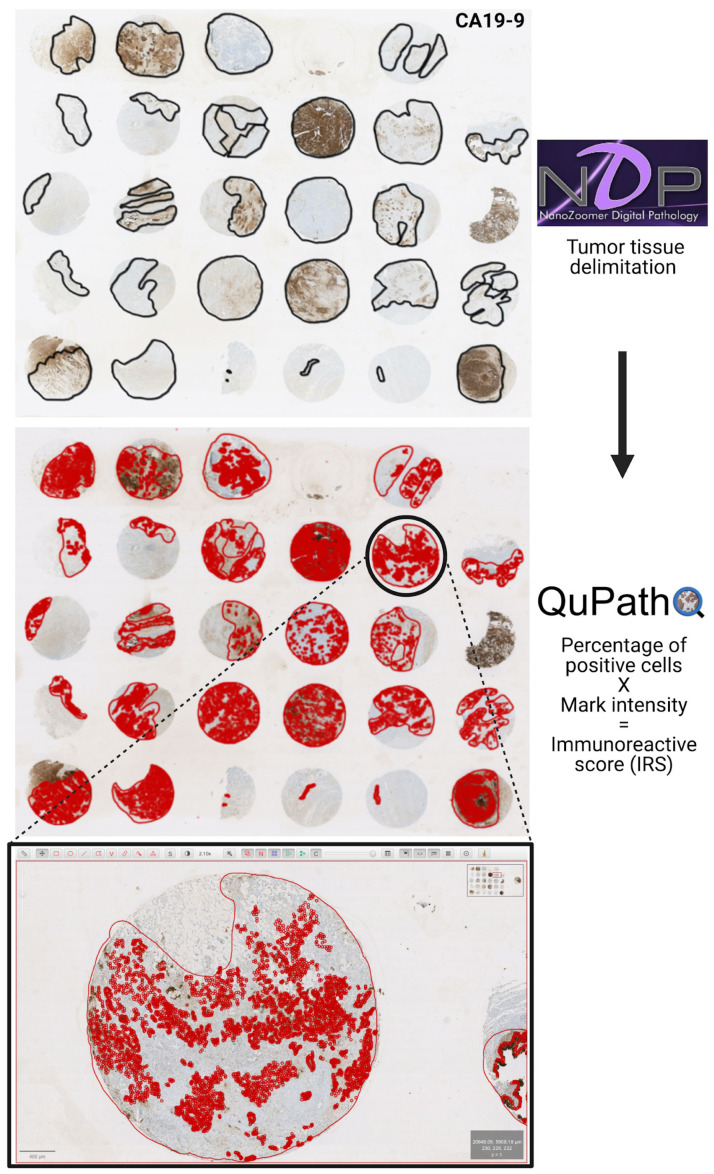
Immunoreactive scores (IRS) determination workflow. First, tumor tissue delimitations were performed using NDP-viewer 2.5 and pathologist expertise. Second, single cells were selected (red) and IRS was calculated by determining the percentage of positive tumor cells and the mark intensity using the QuPath. After automated IHC array staining, digital image analysis was conducted; an array for the CA19-9 marker is shown as an example.

**Figure 2 cancers-15-03440-f002:**
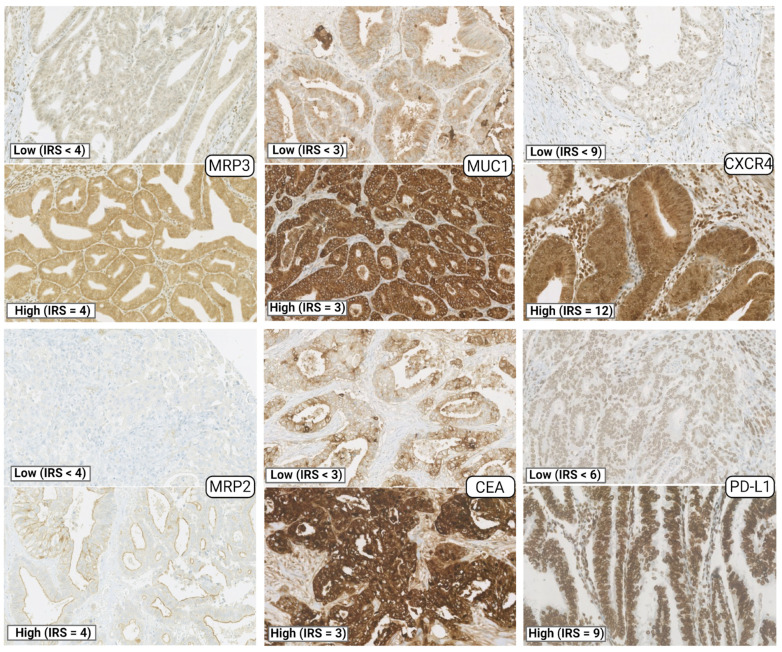
Examples of immunohistochemical images for representative tumor markers. Images correspond to 80× magnification of the microarray photography. Examples for low and high immunoreactive scores (IRS) for each marker are shown.

**Figure 3 cancers-15-03440-f003:**
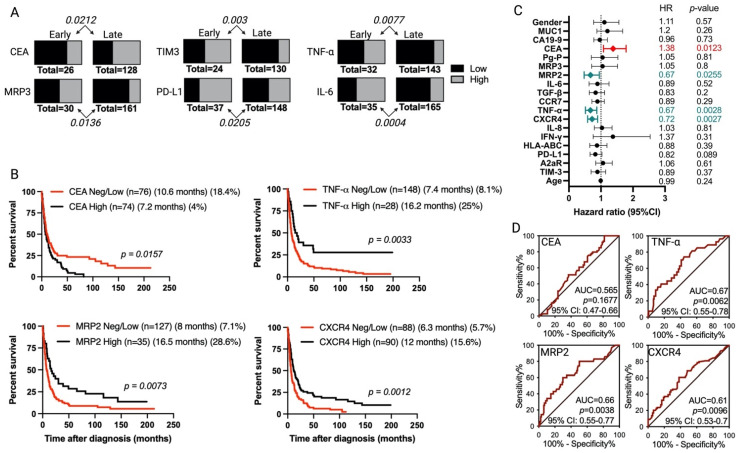
Tumor markers associated with tumor staging and overall survival of gallbladder cancer (GBC) patients. (**A**) A chi-squared test (Yates correction) comparing low (in black)/high (in grey) expression level frequencies between patients with early (TIS + T1) vs. late (T2–T4)-stage tumors. Only statistically significant results are shown. The *p*-values are given above or under the bars. The total number of patients analyzed for each tumor marker is indicated. (**B**) Kaplan–Meier post-diagnosis overall-survival (OS) estimation of late-stage tumor-bearing GBC patients, according to tumor expression patterns (neg/low in red lines vs. high expression in teal lines). Each graph shows the number of patients in each group (n), the median OS time in months, and the 5-year survival rate (%). The figure shows only the markers with statistically significant differences. The *p*-values were calculated using a log-rank (Mantel–Cox) test. (**C**) Univariate Cox regression analyses for GBC patients having late-stage tumors (*n* = 195). The figure shows the actual hazard ratios (HR) and *p*-values on the right. For gender, HR was calculated using the female gender as the reference. (**D**) Receiver operating characteristic (ROC) curves for patient survival, according to the tumor marker pattern expressions. AUC: area under the curve; CI: confidence interval.

**Figure 4 cancers-15-03440-f004:**
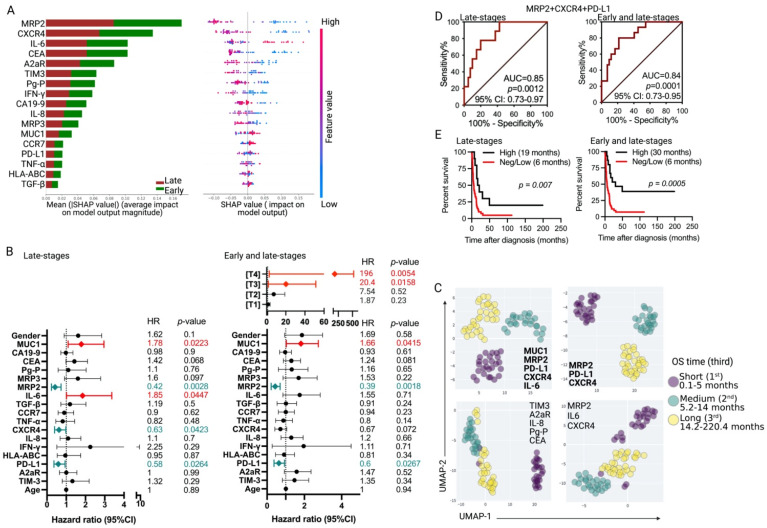
Multivariate analyses identified a group of tumor markers that distinguish between the long-term and short-term survivors of gallbladder cancer (GBC). (**A**) Tumor marker expression (feature) contribution across the samples. The global feature contributions were ranked according to the mean (|Tree SHAP|) across all the samples using a classification model, considering the tumor staging as classes (**left**). This was the same case for the individual feature contributions (**right**). (**B**) Cox multiple regression analyses for the GBC patients with late-stage (T2–T4) tumors (*n* = 195 patients, **left**) and for the complete cohort (*n* = 241 patients, **right**). The figure shows the actual hazard ratios (HR) and *p*-values on the right. HR was calculated using tumor in situ values as a reference for tumor staging, while for gender, HR was calculated using the female values as a reference. Statistically significant HR are highlighted in colors (red: higher hazard; teal: lower hazard). (**C**) Dimensionality reduction analysis using uniform multiple approximation and projection (UMAP) shows that the combination of MUC1, IL-6, MRP2, PD-L1, and CXCR4 expression levels allows a suitable arrangement of GBC patients according to their prognosis, with OS time divided into three groups: short- (0.1–5 months), medium- (5.2–14 months), and long-term (14.2–220.4 months) survival. Then, we searched for the minimal combination of markers capable of keeping the separation of patients according to survival time. Only the cluster MRP2/PD-L1/CXCR4 showed a clear separation of patients. Any other combination of markers was not able to accurately discriminate among patients according to their OS times (some examples are shown at the bottom). (**D**) Receiver operating characteristic (ROC) curve for patient survival according to the combination of MRP2/CXCR4/PD-L1 pattern expressions (negative/low co-expression vs. high co-expression) in late-stage tumor patients (**left**) or the complete cohort (**right**). (**E**) Kaplan–Meier post-diagnosis OS estimation for GBC patients according to MRP2/CXCR4/PD-L1 tumor expression (negative/low co-expression vs. high co-expression) in late-stage tumor patients (**left**) or the complete cohort (**right**). The median OS times are shown in months. AUC: area under the curve; CI: confidence interval; SHAP: Shapley additive explanation.

**Table 1 cancers-15-03440-t001:** Clinicopathological characteristics of gallbladder cancer patients.

Gender	Female	79.2% (*n* = 191)
	Male	20.8% (*n* = 50)
Mean age (SD)	All	65.1 (±11.2)
	Female	64.4 (±11.4)
	Male	67.6 (±10.4)
Tumor differentiation	Poor	30.8% (*n* = 65)
		Median OS time: 6 months
		Five-year survival rate: 3.1%
	Moderate	56.9% (*n* = 120)
		Median OS time: 16.5 months
		Five-year survival rate: 20%
	Well	12.3% (*n* = 26)
		Median OS time: 178.5 months
		Five-year survival rate: 46.1%
Tumor staging	Early stages (TIS + T1)	46 patients (22 TIS + 24 T1)
		41 female, 5 male
		Mean age: 61.6 (±11.4)
		Median OS time: 220 months
		Five-year survival rate: 52.2%
	Late stages (T2 + T3 + T4)	195 patients (52 T2 + 137 T3 + 6 T4)
		150 female, 45 male
		Mean age: 65.9 (±11)
		Median OS time: 10 months
		Five-year survival rate: 11.8%

OS: overall survival; SD: standard deviation; TIS: tumor in situ.

**Table 2 cancers-15-03440-t002:** Association between tumor marker expression and tumor staging in gallbladder cancer patients.

Marker	Number (%) of Positive Cases, Distribution Using IRS	
	Early Stage Tumors (TIS + T1)	Late-Stage Tumors (T2–T4)
*Epithelial tumor markers*		
MUC1	38 (100%): 20 Low + 18 High	171 (98.8%): 60 Low + 111 High
CEA	26 (83.9%): 18 Low + 8 High	128 (85.3%): 54 Low + 74 High
CA19-9	31 (86.1%): 11 Low + 20 High	145 (83.8%): 49 Low + 96 High
*Multidrug or apoptosis resistance*		
MRP2	29 (93.5%): 17 Low + 12 High	159 (98.1%): 124 Low + 35 High
MRP3	30 (100%): 16 Low + 14 High	161 (99.4%): 124 Low + 37 High
Pg-P	28 (100%): 17 Low + 11 High	159 (99.4%): 123 Low + 36 High
Survivin	36 (97.3%)	145 (92.9%)
*Immunological markers*		
TIM3	24 (76%): 11 Low + 13 High	130 (78.8%): 101 Low + 29 High
PD-L1	37 (88.1%): 12 Low + 25 High	148 (82.7%): 82 Low + 66 High
A2aR	39 (88.6%): 27 Low + 12 High	145 (79.7%): 114 Low + 31 High
HLA-ABC	42 (97.7%): 15 Low + 27 High	180 (98.4%): 71 Low + 109 High
IFN-γ	39 (97.5%): 34 Low + 5 High	166 (94.8%): 161 Low + 5 High
IL-8	34 (80.9%): 26 Low + 8 High	152 (84.9%): 119 Low + 33 High
CXCR4	39 (95.1%): 10 Low + 29 High	153 (85.9%): 63 Low + 90 High
TNF-α	32 (76.2%): 18 Low + 14 High	143 (81.2%): 115 Low + 28 High
CCR7	17 (65.4%): 9 Low + 8 High	105 (66.4%): 59 Low + 46 High
TGF-β	39 (97.5%): 6 Low + 33 High	169 (97.1%): 56 Low + 113 High
IL-6	35 (100%): 14 Low + 21 High	165 (97.6%): 120 Low + 45 High

IRS: immunoreactive score.

**Table 3 cancers-15-03440-t003:** Clinicopathological characteristics of the gallbladder cancer patients with high and neg/low expression of the cluster MRP2/CXCR4/PD-L1.

Complete Cohort	MRP2/CXCR4/PD-L1 Co-Expression Pattern		*p*-Value
	High (*n* = 15)	Neg/Low (*n* = 42)	
Mean age (SD)	66.3 (±7.9)	64.7 (±11.3)	0.62 ^a^
Gender (F/M)	12/3	54/8	0.48 ^b^
*Tumor differentiation*			
Poor	15.4% (*n* = 2)	39% (*n* = 16)	0.11 ^b^
Moderate	61.5% (*n* = 8)	48.8% (*n* = 20)	
Well	23.1% (*n* = 3)	12.2% (*n* = 5)	
*Primary tumor staging (TNM)*			
Early stages (TIS + T1)	33.3% (*n* = 5)	4.8% (*n* = 2)	0.0038 ^b^
Late stages (T2–T4)	66.7% (*n* = 10)	95.2% (*n* = 40)	
*Cancer disease stage*			
I	33.3% (*n* = 5)	4.8% (*n* = 2)	0.0627 ^b^
II	26.7% (*n* = 4)	19% (*n* = 8)	
III	26. 7% (*n* = 4)	64.3% (*n* = 27)	
IV	13.3% (*n* = 2)	11.9% (*n* = 5)	
*R status*			
R0	66.7% (*n* = 10)	45.2% (*n* = 19)	0.38 ^b^
R1	26.7% (*n* = 4)	40.5% (*n* = 17)	
R2	6.6% (*n* = 1)	14.3% (*n* = 6)	
**Late-stage (T2–T4) tumors**	**High (*n* = 10)**	**Neg/Low (*n* = 40)**	
Mean age (SD)	64.3 (±8)	65.1 (±11.2)	0.83 ^a^
Gender (F/M)	8/2	54/8	0.55 ^b^
*Tumor differentiation*			
Poor	20% (*n* = 2)	39% (*n* = 16)	0.26 ^b^
Moderate	60% (*n* = 6)	48.8% (*n* = 20)	
Well	20% (*n* = 2)	12.2% (*n* = 5)	
*Cancer disease stage*			
I	0% (*n* = 0)	0% (*n* = 0)	0.36 ^b^
II	40% (*n* = 4)	20% (*n* = 8)	
III	40% (*n* = 4)	67.5% (*n* = 27)	
IV	20% (*n* = 2)	12.5% (*n* = 5)	
*R status*			
R0	50% (*n* = 5)	42.5% (*n* = 17)	0.77 ^b^
R1	40% (*n* = 4)	42.5% (*n* = 17)	
R2	10% (*n* = 1)	15% (*n* = 6)	

SD: standard deviation; F: female; M: male. ^a^ T test; ^b^ χ^2^ test (Yates correction).

## Data Availability

Available under reasonable request.

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
