# Peer review of "Co-Expression of Immunohistochemical Markers MRP2, CXCR4, and PD-L1 in Gallbladder Tumors Is Associated with Prolonged Patient Survival"

_cancers, 2023, doi:10.3390/cancers15133440_

Round 1
Reviewer 1 Report (Previous Reviewer 1)
The authors have revised their manuscript based on the reviewers’ suggestions. My remaining comments are as follows.
1. “Late-stage” generally refers to cancer stage (such as III or IV) rather than tumor size (such as T3 or T4). Because the authors have kindly added cancer stage, the authors should clarify which they are talking about in each instance of the word “late-stage”, to avoid confusing the reader.
2. Figure 3C shows univariate Cox analysis, and multivariate analysis is explained from line 360 onward. Why are variables that were not significant in univariate analysis, such as MUC1 and PD-1, included in multivariate analysis? Please explain in Methods how items were selected for the multivariate analysis.
3. Figure 4B looks very much like Figure 3C, but in 4B the authors are showing results for multivariate analysis. This is very misleading to the reader. To avoid confusion, please least HRs and p-values for all variables in the univariate analysis and all included variables in the multivariate analysis. Again, all variables in Figure 4B are included in multivariate analysis, even through some of them were probably not significant in univariate analysis.
There are still many English errors throughout the manuscript.
Author Response
Reviewer 1:
"Late-stage" generally refers to cancer stage (such as III or IV) rather than tumor size (such as T3 or T4). Because the authors have kindly added cancer stage, the authors should clarify which they are talking about in each instance of the word "late-stage", to avoid confusing the reader.
We highly appreciate the reviewer's appointment regarding stage classification.
In our manuscript we refer to tumor samples obtained at the surgery timepoint, we classified the patient samples by T-stage classification that consider the size and spread of the primary tumor according to the AJCC/UICC, TNM Classification 2010. We categorized the samples into two groups: patients with early-stage tumors (TIS + T1) and patients with late-stage tumors (T2. T3, and T4). Furthermore, we changed the enunciation "Early-stage patients" for "patients with early-stage primary tumor" in the text.
To ensure clarity, we have highlighted this point in the manuscript:
Line 119: "Primary tumor staging was established on the reported imaging findings by a pathologist evaluating the resected specimens. GBC samples were classified into two categories: primary tumors in early-stage defined by adenocarcinoma confined to the tunica muscularis [tumor in situ (TIS), T1a, or T1b] or late-stage tumors encompassed T2, T3, and T4, indicating more advanced tumor progression according to the eighth edition of the American Joint Committee on Cancer (AJCC) Staging Manual and the AJCC/UICC TNM Classification 2010 [25]."
Line 240: "Using T-stages that consider the size and spread of the tumor according to the AJCC/UICC, TNM Classification 2010, we categorized the samples into two groups: patients with early-stage tumors (TIS + T1) and patients with late-stage tumors (T2. T3, and T4). Among included patients, we observed 22 (9%) with tumor in situ (TIS), 24 (10%) with T1, 52 (22%) with T2, 137 (57%) with T3, and 6 (2%) with T4.
Additionally, to further support the understanding of the categorization, we have included the respective categories of early and late tumors in Tables 2 and 3, as well as in the legends of Figures 3 and 4.
Figure 3C shows univariate Cox analysis, and multivariate analysis is explained from line 360 onward. Why are variables that were not significant in univariate analysis, such as MUC1 and PD-1, included in multivariate analysis? Please explain in Methods how items were selected for the multivariate analysis.
As explained below all the variables were analyzed in univariate and multivariate analysis without selection.
Figure 4B looks very much like Figure 3C, but in 4B the authors are showing results for multivariate analysis. This is very misleading to the reader. To avoid confusion, please least HRs and p-values for all variables in the univariate analysis and all included variables in the multivariate analysis. Again, all variables in Figure 4B are included in multivariate analysis, even through some of them were probably not significant in univariate analysis.
We thank the Reviewer for the critical comment and we apologize for the confusion. We would like to clarify that as the reviewer mention we included all the markers values, significant or not in the multivariate analysis. We wanted to explore the possibility that certain variables, which may not show significance in univariate analysis, could still exhibit significance in a multivariate analysis. Therefore, in our Cox multiple regression analysis, we included all the variables used in the univariate Cox regressions, even those that did not show a significant association with overall survival.
There are several potential scenarios that could explain why some variables become significant in multivariate analysis despite not being significant in univariate analysis: (1) unbalanced sample sizes, (2) influence of missing data, (3) large within-group variation relative to between-group variation, and (4) presence of variable interactions. This phenomenon has been documented by Lo et al. [Changgeng Yi Xue Za Zhi. 1995;18(2):95-101] and others. Although we cannot determine which of these scenarios precisely applies to our data sets, our findings align with the overarching conclusion. Notably, MUC1, IL-6, and PD-L1 demonstrated significance in multiple Cox regression analyses but not in univariate Cox analysis.
To clarify our approach to the readers, we have included the following sentence in the Methods section:
“Univariate and multivariate analyses were performed using the Cox proportional-hazard regression model (Breslow method) to study the effects of different variables on OS. All variables included in the univariate regression analysis were considered in the multiple regression analysis except survivin, which exhibited homogeneous staining among tumor samples.”
Additionally, we added the HR and p-values obtained for tumor staging variables in the multiple Cox regression analysis (Fig 4B, right), to show all the variables used in the analysis.
Comments on the Quality of English Language
There are still many English errors throughout the manuscript.
We are revised all the manuscript and corrected the language errors.

Reviewer 2 Report (Previous Reviewer 2)
The authors addressed previous concerns.
Author Response
We appreciate the reviewer comments.
Round 2
Reviewer 1 Report (Previous Reviewer 1)
The authors have revised their manuscript based on my suggestions. I have no further suggestions.
Their English language revisions were not marked and I could not confirm them all, but the manuscript appears to be improved.
This manuscript is a resubmission of an earlier submission. The following is a list of the peer review reports and author responses from that submission.
Round 1
Reviewer 1 Report
The authors analyze IHC markers in gallbladder cancer patients. This is a potentially interesting study but baseline data is lacking and the manuscript is hard to follow. My concerns are as follows.
Major:
1. Materials and methods: Details on inclusion criteria and exclusion criteria are necessary.
2. The authors state that they obtained cholecystectomy specimens, but then go on to say the collected 253 biopsies from 241 patients. I do not understand this; patients only have one gallbladder.
3. I assume this was a retrospective study. However specimens from 2000 and 2019 are included and the approval date was December 2015. Please clarify.
4. More baseline characteristics are required in Table 1. Variables that affect overall survival must be included: performance status, stage (not just T stage but stage I-IV), serum tumor markers (especially CA19-9), type of surgery (e.g. was lymph node dissection performed?), whether R0 resection was achieved, whether adjuvant or palliative chemotherapy was received, recurrence, etc. Also T2-4 should not be lumped together.
5. How long was the follow up period?
6. A table with actual figures showing hazard ratios and p-values is desirable.
7. It is unclear how the MRP2/CXCR4/PD-L1 combination was calculated. Are the authors comparing cases with all 3 positive vs. those with all 3 negative? Please clarify in the manuscript. If this is the main conclusion, much more detail and discussion is required.
8. It is unclear whether the 3 markers are just markers for early stage disease (which is better probably evaluated on CT) or if they are predictors of favorable OS regardless of cancer stage. Also, how would these markers change clinical practice?
9. Was PD-L1 associated with better survival because immunotherapy was used in some patients?
Minor:
1. Title: I do not believe a reference to “multivariate analysis” should be included in the title.
2. Introduction: I do not believe an explanation of multivariate analysis is necessary.
3. Figure 1 is meaningless; it is obvious that early stage GBC has better prognosis than late stage GBC.
Author Response
We would like to thank the referees for the thorough review of our paper and their suggestions, which have improved the current version of our manuscript. The point-by-point answers to the reviewers’ comments (in BOLD) are as follows:
- Materials and methods: Details on inclusion criteria and exclusion criteria are necessary.
R: In section 2.1, we include more information for the inclusion and exclusion criteria and for the analyzed samples. “The selection criteria of the biopsies were based on sample etiology; thus, only invasive primary gallbladder adenocarcinomas were selected. The pathologist excluded adenocarcinoma, squamous carcinoma, neuroendocrine carcinoma, tumor-infiltrated lymph nodes, and metastases.”
- The authors state that they obtained cholecystectomy specimens, but then go on to say the collected 253 biopsies from 241 patients. I do not understand this; patients only have one gallbladder.
R: We appreciate that the reviewer highlighted this mistake. We collected samples from 253 patients, but only 241 fulfilled the criteria of sample integrity and quality and were analyzed, but as this is irrelevant and, to not confuse, we changed the sentence to: “According to these criteria, it was possible to collect 241 biopsies,…”
- I assume this was a retrospective study. However, specimens from 2000 and 2019 are included and the approval date was December 2015. Please clarify.
R: We agree with the reviewer. It is a retrospective study (now pointed out in line 125). As we described in the “Institutional Review Board Statement” and now also in section 2.1, “the Bioethical Committee for Human Research of the Valdivia Regional Hospital (protocol code 403, date of approval, December 2015) and the Universidad de Chile Ethics Committee for Studies Involving Humans (code 086-2017, date of approval July 2017) approved the retrospective study, allowing the analysis of samples from the year 2000. The protocol was revised again by the Bioethical Committee for Human Research of the Valdivia Regional Hospital (code 082-2020, date of approval March 2020). We were authorized to include new samples until 2019.
- More baseline characteristics are required in Table 1. Variables that affect overall survival must be included: performance status, stage (not just T stage but stage I-IV), serum tumor markers (especially CA19-9), type of surgery (e.g. was lymph node dissection performed?), whether R0 resection was achieved, whether adjuvant or palliative chemotherapy was received, recurrence, etc. Also T2-4 should not be lumped together.
R: We agree with the reviewer that multiple clinical factors, such as the treatments received and the general physical status of the patients, impact overall survival. Unfortunately, these differences go far beyond the elements proposed by the reviewer for the analysis, which, while relevant, will never be comprehensive. In this study, we wanted to determine, as a first approach, the molecular markers present in the tumor associated with the survival of a heterogeneous and representative group of patients with gallbladder cancer. All the patients come from the same hospital center and have received the standard treatments for these cases, which are very limited and do not include new technologies such as immunotherapies. For this reason, we included variables associated mainly with the characteristics of the biopsies and those clinical characteristics that cover the entire cohort of patients.
However, as suggested by the reviewer, we include a new parameter related to tumor differentiation status (poor, moderate, and well differentiated) in Table 1 and Section 3.1.
Respect to the comment “T2-4 should not be lumped together”, we included this sentence in methods section to base our classification: “The early GBC stage patients were defined by the presence of adenocarcinoma confined to the tunica muscularis (Tis, T1a, or T1b) according the AJCC/UICC, TNM Classification 2010”, as we previously published (doi.org/10.1186/s12885-018-4147-6).
- How long was the follow up period?
R: We include in section 3.1 the next sentence: “The median follow-up period was 145 months for the complete cohort; being not statistically different between patients with early-stage (99.2 months) and late-stage tumors (146 months) (p = 0.47)”.
- A table with actual figures showing hazard ratios and p-values is desirable.
R: The values for HR and p-values for each marker are now shown at the right of each figure (new Figures 3 and 4) as suggested by the reviewer. Figure legends were changed accordingly.
- It is unclear how the MRP2/CXCR4/PD-L1 combination was calculated. Are the authors comparing cases with all 3 positive vs. those with all 3 negative? Please clarify in the manuscript. If this is the main conclusion, much more detail and discussion is required.
R: We thank the reviewer for pointing out this issue. We first performed univariate and then multivariate analysis of markers expression and patient survival. From those emerged a group of markers that showed correlations. The definition of high and low expression for each marker is described in the methods section. The global feature contributions of each marker were ranked according to the mean (|Tree SHAP|) across all the samples using a classification model considering the tumor staging as classes. According to hazard ratios, five markers showed positive or negative significant correlations with patient’s overall survival (MUC1/IL-6/MRP2/CXCR4/PD-L1). Please note that for UMAP analysis, the expression data is evaluated without pre-categorize them in low or high. In the current version of the manuscript, we provide more details about the used methodology. Then, we search for the minimal combination of markers capable of keeping the separation of patients according to survival time. Only the cluster MRP2/PD-L1/CXCR4 showed a clear separation of patients. Section 3.3: “Based on these results, we further performed ROC curve analysis on patients with late-stage tumors, showing a good prognostic value for high co-expression of MRP2, CXCR4, and PD-L1 (n = 10 patients) as compared with negative/low co-expression (n = 40 patients) of these markers (AUC = 0.85, p = 0.0012) (Figure 4D, left). Consistently, high co-expression levels of the 3 markers combined were strongly associated with in-creased OS times as compared with its negative/low co-expression (median OS 6 vs. 19 months, p = 0.007) (Figure 4E, left).” Figure 4 legend also changed accordingly.
Moreover, we include in discussion the next line: “We found that among our late-stage tumor cohort, 40 patients meet the criteria of having negative/low tumor co-expression of MRP2/CXCR4/PD-L1, which according to our results, it translates into a very poor prognosis (HR = 2.62; median OS time of only 6 months). Moreover, high co-expression of MRP2/CXCR4/PD-L1 could have a prognostic value for GBC regardless of tumor staging (right panels of Figures 4D and E)”.
- It is unclear whether the 3 markers are just markers for early stage disease (which is better probably evaluated on CT) or if they are predictors of favorable OS regardless of cancer stage. Also, how would these markers change clinical practice?
R: We thank the reviewer for appoint this question. As described in section 3, “there was no statistically significant association between the expression levels of the 18 markers and the OS time of patients with early-stage GBC (data not shown). Therefore, unless stated, the results described below were obtained in patients with late-stage tumors”.
Additionally, motivated by the reviewer's question, we went deep into the analysis of this cluster in our complete patient cohort and included these new results in section 3.3: “Among the 46 patients with early-stage tumors, four meet the criteria of having neg/low co-expression of the cluster MRP2/CXCR4/PD-L1, and five of these patients have the opposite profile, which makes it impossible to evaluate its predictor value in patients with early-stage tumors statistically. However, when these nine patients were included in these analyses, we found that MRP2/CXCR4/PD-L1 high co-expression has an excellent predictive value (AUC = 0.81, p = 0.0004; Figure 4D, right) and is a predictor of favorable OS (30 months vs. 6 months; HR = 0.4; p = 0.0025) regardless tumor stage (Figure 4E, right).”
- Was PD-L1 associated with better survival because immunotherapy was used in some patients?
R: The cohort of Chilean GBC patients included in this study did not were treated with immunotherapy.
Minor:
- Title: I do not believe a reference to “multivariate analysis” should be included in the title.
R: The title was changed to: Co-expression of immunohistochemical markers MRP2, CXCR4, and PD-L1 in gallbladder tumors is associated with patients prolonged survival
- Introduction: I do not believe an explanation of multivariate analysis is necessary.
R: The explanation and references associated were removed.
- Figure 1 is meaningless; it is obvious that early stage GBC has better prognosis than late stage GBC.
R: Figure 1 was eliminated, and the data were included in the new Table 1, in order to corroborate that the studied cohort is representative of what is reported in the literature.

Reviewer 2 Report
The authors examined associations between the expression of various epithelial, multidrug and apoptosis resistance, and immunologic markers and patients’ prognoses in 241 primary gallbladder adenocarcinomas to know molecules having impacts on the disease prognosis and management. They found that a significant association between MRP2 (p = 0.0028), CXCR4 (p = 0.0423), or PD-L1 (p = 0.0264) expression and better prognosis for patients with late-stage tumors. The expression of the MRP2/CXCR4/PD-L1 cluster of markers discriminates among short, medium, and long-term patient survival, with a ROC significant prognostic value (AUC = 0.85, p = 0.0012). The high expression of combined markers is associated with increased survival times (p = 0.007). Conclusion was made as follows: These results suggest that the MRP2/CXCR4/PD-L1 cluster can potentially be prognostic markers for GBC. This study seems to be conducted in a careful objective manner with good number of patients’ samples. Following concerns should be addressed:
1. The results obtained in this study seem to be controversial according to previously published results elsewhere. The MRP2, CXCR4, and PD-L1 have been reported to be associated with poor prognosis of patients with various cancers when they are expressed in high level. However, this study shows that the low expression of these molecules was associated with poor prognosis in patients with late stage of GBC. The authors need to provide more detailed interpretation for these controversial results with citing more papers. For example, regarding PD-L1, the authors only discussed about reports indicating the association of PD-L1 expression in tumor cells and better prognosis, however, a large number of reports indeed indicate high PD-L1 expression in tumors and poor prognosis of patients.
2. The authors divided patients into two cohorts of those of early stage and late stage for analysis of associations between the expression of molecules and patients’ prognosis. Why the association was not analyzed in entire patients at first? Such analysis should be conducted and reported even without any significance. Or the authors need to provide a reasonably excuse to ignore such analysis.
3. For evaluation of IRS, how the percentage of positive cells was divided into 5 categories? Was it just evenly divided, i.e., 20% threshold?
4. The authors need to address why some markers were evaluated only by intensity, or others by the combination of intensity and proportion.
5. Why samples with IRS = 0 were excluded from analysis?
6. What were thresholds of short, medium, and long-term survival described in line 359?
7. Figure 5C is highly interesting, however, how the graph can be read is obscure. What kind of the combination of the expression of MRP2/PD-L1/CXCR4 was shown in the short-term survival patients compared with medium or long-termed survival patients? Can this be interpreted that the short, medium, and long-term survival patients showed low expression of all, two, and one the three markers, respectively, or was there any specific pattern of expression?
Author Response
We would like to thank the referees for the thorough review of our paper and their suggestions, which have improved the current version of our manuscript. The point-by-point answers to the reviewers’ comments (in BOLD) are as follows:
- The results obtained in this study seem to be controversial according to previously published results elsewhere. The MRP2, CXCR4, and PD-L1 have been reported to be associated with poor prognosis of patients with various cancers when they are expressed in high level. However, this study shows that the low expression of these molecules was associated with poor prognosis in patients with late stage of GBC. The authors need to provide more detailed interpretation for these controversial results with citing more papers. For example, regarding PD-L1, the authors only discussed about reports indicating the association of PD-L1 expression in tumor cells and better prognosis, however, a large number of reports indeed indicate high PD-L1 expression in tumors and poor prognosis of patients.
R: We agree with the reviewer. In a new version of the manuscript, we provide more speculative interpretation of the results in the discussion section.
“A possible interpretation of this potentially controversial result is that in our cohort, the high positivity rate of MRP2, could indicate a physiological role of MRP2, which is supporting terminal excretion and detoxification of endogenous organic anions and up-take/excretion of bile salts in the absence of chemotherapy [48]. Is worth to be noted that the patients included in our study had not undergone any pre-operative chemotherapy.”
“Although our findings contradict numerous reports [reviewed in 50], there are consistent with a study demonstrating CXCR4 expression as a favorable prognostic indica-tor for survival in multiple myeloma patients [52]. These controversial results are congruent with the fact that clinical inhibition of the CXCL12-CXCR4 axis using FDA-approved inhibitors such as AMD3100 or Plerixafor has yielded inconsistent outcomes, contrary to promising preclinical findings [50].
High expression of CXCR4 in the cytoplasm or membrane of tumor cells is often a predictor of poor prognosis. In contrast, high expression in the nucleus is typically associated with a better prognosis. For example, in breast cancer, CXCR4 was found in both the cytoplasm and nuclei of tumor cells, but only cytoplasmic expression was linked to lymph node metastasis [53]. In this context, Su et al. [54] and Wagner et al. [55] discovered that plasma membrane expression of CXCR4 in lung adenocarcinoma is an independent risk factor associated with poor disease-free survival. In contrast, nuclear expression confers a survival benefit. Therefore, CXCR4 may promote tumor cell proliferation and metastasis when present in the cytoplasm or cell membrane. Still, its effects are likely to be less pronounced when it is localized in the nucleus. Although not analyzed, these antecedents suggest that CXCR4 could be preferentially accumulated at the nucleus of tumor cells in our cohort. Further research and clinical trials are warrant-ed to understand better the complex role of the CXCR4 (particularly the expression of the ligand-activated p S339 form of CXCR4) in cancer progression and to explore potential strategies for effective therapeutic targeting of this axis in different tumor types, including GBC.”
“Our univariate analysis of PD-L1 is consistent with these findings; however, we observed a potential association with prognosis when analyzed jointly with CXCR4 and MRP2 co-expression (Figures 4 C-E). Of note, many reports have also shown an association between PD-L1 expression and poor prognosis in patients with different solid tumors, including hepatocellular carcinoma [59], breast cancer [60], and oral squamous cell carcinoma [61]. The differential immune landscape status of the different cohorts studied could explain this controversy.”
- The authors divided patients into two cohorts of those of early stage and late stage for analysis of associations between the expression of molecules and patients’ prognosis. Why the association was not analyzed in entire patients at first? Such analysis should be conducted and reported even without any significance. Or the authors need to provide a reasonably excuse to ignore such analysis.
R: We thank the reviewer for pointing out this issue. We now include a multivariate Cox regression analysis for the complete cohort in section 3.3 (new Figure 4B, left panel): " Analysis in the complete cohort shows that MRP2 (HR = 0.39, 95% CI = 0.2172-0.6992, p = 0.0018) and PD-L1 (HR = 0.6, 95% CI = 0.3739-0.9352, p = 0.0267) are associated with better prognosis, whereas MUC1 expression (HR = 1.66, 95% CI = 1.03-2.751, p = 0.0415) is associated with poor prognosis in GBC patients regardless tumor staging (Figure 4B, right)".
Moreover, motivated by the reviewer's questions, we deeply analyzed this marker cluster in our complete patient cohort. We included these new results in section 3.3: " Among the 46 patients with early-stage tumors, four meet the criteria of having neg/low co-expression of the cluster MRP2/CXCR4/PD-L1, and five of these patients have the opposite profile, which makes it impossible to evaluate its predictor value in patients with early-stage tumors statistically. However, when these nine patients were included in the total group of patients analyzed, we found that MRP2/CXCR4/PD-L1 high co-expression has an excellent predictive value (AUC = 0.81, p = 0.0004; Figure 4D, right) of favorable OS (30 months vs. 6 months; HR = 0.4; p = 0.0025) regardless tumor stage (Figure 4E, right)."
- For evaluation of IRS, how the percentage of positive cells was divided into 5 categories? Was it just evenly divided, i.e., 20% threshold?
R: We clarify how the percentage of positive cells was established, including the following sentences in section 2.5: “For immune-associated markers, the percentage of positive cells in a tumor area was transformed into four (0, 1, 2, 3) or five categorical scores (0, 1, 2, 3, 4), depending on the type of marker. For TIM3 and PD-L1, the percentages of positive cells were divided into four categorical scores, as previously described by Peng et al. (2017) [27] and Neyaz et al. (2018) [28], respectively. For A2aR, the percentages of positive cells were divided into five categorical scores, as described by Wu et al. (2019) [29]. For the rest of the immunological markers, the percentages of positive cells were divided evenly into five categorical scores (0-19%, 20-39%, 40-59%, 60-79%, 80-100%).”
- The authors need to address why some markers were evaluated only by intensity, or others by the combination of intensity and proportion.
R: We thank the reviewer for pointing out this issue. We clarified this point in section 2.5 in the new version of the manuscript as follow: “The non-immune related markers analyzed (MRP2, MRP3, Pg-P, CEA, CA19-9, survivin, and MUC1) shown a homogeneous distribution of expression in the tumor tissue, therefore their IRS only considered the intensity scores.”
- Why samples with IRS = 0 were excluded from analysis?
R: We are sorry for the confusion. Samples with negative expression (IRS = 0) were not excluded from the analysis; they were considered in the “Low” group in Figure 4. Therefore, we changed the results description and new Figure 4 legend to clarify this.
- What were thresholds of short, medium, and long-term survival described in line 359?
R: We thank the reviewer for pointing out this issue. The threshold description for short (0.1-5 months), medium (5.2-14 months), and long-term (14.2-220.4) survival were included in the new Figure 4C.
- Figure 5C is highly interesting, however, how the graph can be read is obscure. What kind of the combination of the expression of MRP2/PD-L1/CXCR4 was shown in the short-term survival patients compared with medium or long-termed survival patients? Can this be interpreted that the short, medium, and long-term survival patients showed low expression of all, two, and one the three markers, respectively, or was there any specific pattern of expression?
R: For UMAP analysis, a machine-learning algorithm was used to detect group of markers (from 1 to 18 markers) whose expression levels (without pre-categorize them in neg, low, or high) could cluster together GBC patients according to their OS times. The algorithm (as t-SNE) is used for dimensionality reduction and data visualization in two dimensions which makes it easier to understand and interpret.
We first performed univariate and then multivariate analysis of markers expression and patient survival. From those emerged a group of markers that showed correlations. The definition of high and low expression for each marker is described in the methods section. The global feature contributions of each marker were ranked according to the mean (|Tree SHAP|) across all the samples using a classification model considering the tumor staging as classes. According to hazard ratios five markers showed positive or negative significant correlations with patient’s survival (MUC1/IL-6/MRP2/CXCR4/PD-L1). Please note that for UMAP analysis, the expression data is evaluated without pre-categorize them in low or high. In the current version of the manuscript, we provide more details about the used methodology. Then, we search for the minimal combination of markers capable of keeping the separation of patients according to survival time. Only the cluster MRP2/PD-L1/CXCR4 showed a clear separation of patients. Section 3.3: “Based on these results, we further performed ROC curve analysis on patients with late-stage tumors, showing a good prognostic value for high co-expression of MRP2, CXCR4, and PD-L1 (n = 10 patients) as compared with negative/low co-expression (n = 40 patients) of these markers (AUC = 0.85, p = 0.0012) (Figure 4D, left). Consistently, high co-expression levels of the 3 markers combined were strongly associated with in-creased OS times as compared with its negative/low co-expression (median OS 6 vs. 19 months, p = 0.007) (Figure 4E, left).” Figure 4 legend also changed accordingly.
Moreover, we include in discussion the next line: “We found that among our late-stage tumor cohort, 40 patients meet the criteria of having negative/low tumor co-expression of MRP2/CXCR4/PD-L1, which according to our results, it translates into a very poor prognosis (HR = 2.62; median OS time of only 6 months). Moreover, high co-expression of MRP2/CXCR4/PD-L1 could have a prognostic value for GBC regardless of tumor staging (right panels of Figures 4D and E)”.
In addition, we rephrase the results section to clarify: “Dimensionality reduction analysis using UMAP shows that patients cluster according to their OS times (short-term, medium-term, and long-term survival) only when the expression levels of the group of markers MUC1/IL-6/MRP2/CXCR4/PD-L1 are used as variables. Any other random combination of 5 tumor markers did not allow discriminating GBC patients according to their OS times (Figure 4C).”
Then with the ROC and Kaplan Meier analysis we shown that neg/low co-expression of MRP2/CXCR4/PD-L1 is associated with shorted median OS times, whereas high co-expression of MRP2/CXCR4/PD-L1 is associated with longer median OS times.

Round 2
Reviewer 1 Report
The authors have significantly improved their manuscript based on reviewers’ suggestions. My remaining comments are as follows.
Major:
1. Thank you for providing the new Table 1. Unfortunately, I am not convinced that the authors can conclude that MRP2/CXCR4/PD-L1 co-expression is associated with longer OS based on the data given. Key baseline characteristics are necessary to show that there were no major confounders. If the authors do not have this information, they should at least provide a table to show baseline characteristics (at least including those provided in Table 1) of the MRP2/CXCR4/PD-L1 high co-expression (n=10) and MRP2/CXCR4/PD-L1 low co-expression (n=40) subgroups and provide p-values to show if there are any significant differences between the two groups. Also, I agree that there are countless variables that can affect OS, but that does not mean the authors should not try to eliminate confounding to the extent possible. The major bases should be covered.
Minor:
1. The authors state in Line 117: “The selection criteria of the biopsies were based on sample etiology; thus, only invasive primary gallbladder adenocarcinomas were selected. The pathologist excluded adenocarcinoma, squamous carcinoma, neuroendocrine carcinoma, tumor-infiltrated lymph nodes, and metastases.” The pathologist excluded adenocarcinoma?? Please fix.
2. Table 1: The authors provide median and standard deviation. For age. I would use mean and SD or median and IQR (or range) depending on whether or not the distribution is normal.
Minor English errors are found throughout the manuscript. In particular, the title: “patients prolonged survival” should be revised to “prolonged patient survival”.
Author Response
We would like to thank the referee for the thorough review of our paper and their suggestions, which
have improved the current version of our manuscript. The point-by-point answers to the reviewer new comments (in BOLD) are as follows:
Reviewer 1 Round 2
The authors have significantly improved their manuscript based on reviewers’ suggestions. My remaining comments are as follows.
Major:
- Thank you for providing the new Table 1. Unfortunately, I am not convinced that the authors can conclude that MRP2/CXCR4/PD-L1 co-expression is associated with longer OS based on the data given. Key baseline characteristics are necessary to show that there were no major confounders. If the authors do not have this information, they should at least provide a table to show baseline characteristics (at least including those provided in Table 1) of the MRP2/CXCR4/PD-L1 high co-expression (n=10) and MRP2/CXCR4/PD-L1 low co-expression (n=40) subgroups and provide p-values to show if there are any significant differences between the two groups. Also, I agree that there are countless variables that can affect OS, but that does not mean the authors should not try to eliminate confounding to the extent possible. The major bases should be covered.
R: We understand the reviewer's worry concerning key baseline characteristics affecting association with patient survival markers. Lamentably, access to the individual row clinical data is a complicated process that may involve additional human and material resources, which will not completely solve the apprehensions. According to the pathologist, almost all operations occurred in symptomatic patients, where the tumor was clinically evident and usually advanced, often metastatic and rarely completely resectable. In nearly 90% of cases, resection contained macroscopic residual tumor R2. Moreover, post-operatory treatments for gallbladder patients in this cohort (from the same hospital) do not include adjuvant therapies, and advanced cancer patients receive gemcitabine and cisplatin as gold-standard treatment. The highly homogeneity of the clinical situation may difficult discriminating subgroups that may impact obtained results. However, we will consider the reviewer's suggestion in future prospective studies in new cohorts to confirm the results obtained here.
Following the reviewer's suggestion, we generate Table 3, showing baseline characteristics (those provided in Table 1) for the MRP2/CXCR4/PD-L1 high co-expression and MRP2/CXCR4/PD-L1 low co-expression subgroups (separately by the complete cohort and late-stages tumor patients), providing the p-values. As we described: " There were no statistically significant differences in the mean age, gender distribution, and tumor differentiation among patients with high co-expression and neg/low co-expression of MRP2/CXCR4/PD-L1 (Table 3). However, the group with high co-expression of MRP2/CXCR4/PD-L1 exhibited a higher proportion of early-stage tumors compared to the group with neg/low co-expression (p = 0.0038; Table 3). Therefore, it is important to consider tumor staging as a potential confounding variable that could in-fluence the interpretation of the impact of MRP2/CXCR4/PD-L1 co-expression in the overall patient cohort. However, among patients with late-stage tumors, there were no significant differences in the baseline variables between those with high co-expression and neg/low co-expression of MRP2/CXCR4/PD-L1 (Table 3)".
Minor:
- The authors state in Line 117: “The selection criteria of the biopsies were based on sample etiology; thus, only invasive primary gallbladder adenocarcinomas were selected. The pathologist excluded adenocarcinoma, squamous carcinoma, neuroendocrine carcinoma, tumor-infiltrated lymph nodes, and metastases.” The pathologist excluded adenocarcinoma?? Please fix.
R: We appreciate that the reviewer highlighted this typing mistake. It was corrected: “The pathologist excluded squamous carcinoma, neuroendocrine carcinoma, tumor-infiltrated lymph nodes, and metastases.”
- Table 1: The authors provide median and standard deviation. For age. I would use mean and SD or median and IQR (or range) depending on whether or not the distribution is normal.
R: We changed statistical analysis of age as suggested by the reviewer. According to this analysis, the mean age of patients with early-stage tumors (TIS + T1) was slightly but significantly lower from those with late-stage (T2-T4) tumors (61.6 ± 11.4 vs. 65.9 ± 11; p = 0.0181).
Comments on the Quality of English Language
Minor English errors are found throughout the manuscript. In particular, the title: “patients prolonged survival” should be revised to “prolonged patient survival”.
R: We appreciate the observation. Title was changed. English was revised as suggested by the referee.

Round 3
Reviewer 1 Report
Thank you for adding Table 3, which is helpful.
I was very surprised that almost 90% ended up with R2 resection. It is difficult to believe that all patients were indicated for surgery (were they?). Because 19% of the population had TIS or T1 cancers, this would mean that some of these early stage patients also had R2 resection?? Please explain. Also, this must be mentioned in the manuscript, as most readers would believe that R0 resection was achieved in most, if not all cholecystectomies. I believe this also means that most patients underwent chemotherapy with gemcitabine and cisplatin. Such facts should also be mentioned in the manuscript, as it is important to the reader to understand the study population and factors affecting OS. Exact figures should be provided if available. If baseline characteristics are unavailable, this should be stated as a study limitation.
English language has been improved for the most part.
Author Response
We would like to thank the referee for the thorough review of our paper and their suggestions, which have improved the current version of our manuscript. The point-by-point answers to the reviewer new comments (in BOLD) are as follows:
Reviewer 1 Round 3
I was very surprised that almost 90% ended up with R2 resection. It is difficult to believe that all patients were indicated for surgery (were they?). Because 19% of the population had TIS or T1 cancers, this would mean that some of these early stage patients also had R2 resection?? Please explain.
Also, this must be mentioned in the manuscript, as most readers would believe that R0 resection was achieved in most, if not all cholecystectomies. I believe this also means that most patients underwent chemotherapy with gemcitabine and cisplatin. Such facts should also be mentioned in the manuscript, as it is important to the reader to understand the study population and factors affecting OS. Exact figures should be provided if available. If baseline characteristics are unavailable, this should be stated as a study limitation.
R: We highly appreciate the reviewer's appointment regarding the R stage, and we genuinely apologize for providing an incorrect response to the question about R status in our early letter. The given information in our previous communication was based on a misunderstanding among the clinical pathologists and corresponding authors, which led to incorrect information.
After the reviewer's warning, we performed a thorough analysis of the baseline data from the clinical pathology records of all 57 patients showing MRP2/CXCR4/PD-L1 co-expression pattern differences (Table 3), considering the individual residual tumor (R) classification and patient's disease stage (12.3% with R2, precisely the opposite that previously informed, and in disease stage IV).
The patients were categorized based on high versus negative/low MRP2/CXCR4/PD-L1 co-expression.
The updated Table 3 showed no significant statistical differences between these two groups of patients based on the new baseline variables. The only associated factor was tumor staging (early vs late-stages), as previously mentioned in the manuscript.
Thus, the added baseline variables are not confounding factors that could influence the interpretation of the impact of MRP2/CXCR4/PD-L1 co-expression in our cohort.
Furthermore, even when we excluded patients with R2 status or stage IV from the analysis, the co-expression of MRP2/CXCR4/PD-L1 still exhibited a significant prognostic association, as illustrated in Figures R1 A and R1 B, respectively (enclosed).
